# Tent versus Mask-On Acute Effects during Repeated-Sprint Training in Normobaric Hypoxia and Normoxia

**DOI:** 10.3390/jcm10214879

**Published:** 2021-10-22

**Authors:** Aldo A. Vasquez-Bonilla, Daniel Rojas-Valverde, Adrián González-Custodio, Rafael Timón, Guillermo Olcina

**Affiliations:** 1Grupo de Avances en Entrenamiento Deportivo y Acondicionamiento Físico (GAEDAF), Facultad de Ciencias del Deprote, Universidad de Extremadura, 10003 Cáceres, Spain; adri13agc@gmail.com (A.G.-C.); rtimon@unex.es (R.T.); 2Centro de Investigación y Diagnóstico en Salud y Deporte (CIDISAD), Escuela de Ciencias del Movimiento Humano y Calidad de Vida, Universidad Nacional, Heredia 86-3000, Costa Rica; 3Clínica de Lesiones Deportivas (Rehab&Readapt), Escuela de Ciencias del Movimiento Humano y Calidad de Vida, Universidad Nacional, Heredia 86-3000, Costa Rica

**Keywords:** hypercapnia, cycling, oxygen saturation, CO_2_, endurance, hypoxemia, carbon dioxide, hypercarbia, toxicity

## Abstract

Repeated sprint in hypoxia (RSH) is used to improve supramaximal cycling capacity, but little is known about the potential differences between different systems for creating normobaric hypoxia, such as a chamber, tent, or mask. This study aimed to compare the environmental (carbon dioxide (CO_2_) and wet-globe bulb temperature (WGBT)), perceptual (pain, respiratory difficulty, and rate of perceived exertion (RPE)), and external (peak and mean power output) and internal (peak heart rate (HRpeak), muscle oxygen saturation (SmO_2_), arterial oxygen saturation (SpO_2_), blood lactate and glucose) workload acute effects of an RSH session when performed inside a tent versus using a mask. Twelve well-trained cyclists (age = 29 ± 9.8 years, VO_2_max = 70.3 ± 5.9 mL/kg/min) participated in this single-blind, randomized, crossover trial. Participants completed four sessions of three sets of five repetitions × 10 s:20 s (180 s rest between series) of all-out in different conditions: normoxia in a tent (RSNTent) and mask-on (RSNMask), and normobaric hypoxia in a tent (RSHTent) and mask-on (RSHMask). CO_2_ and WGBT levels increased steadily in all conditions (*p* < 0.01) and were lower when using a mask (RSNMask and RSHMask) than when inside a tent (RSHTent and RSNTent) (*p* < 0.01). RSHTent presented lower SpO_2_ than the other three conditions (*p* < 0.05), and hypoxic conditions presented lower SpO_2_ than normoxic ones (*p* < 0.05). HRpeak, RPE, blood lactate, and blood glucose increased throughout the training, as expected. RSH could lead to acute conditions such as hypoxemia, which may be exacerbated when using a tent to simulate hypoxia compared to a mask-based system.

## 1. Introduction

In elite sport, the performance gap between athletes is getting smaller, and the competition is becoming more intense. For this reason, high-intensity training has become essential in several sports to gain a competitive advantage. It is considered one of the most effective strategies for enhancing performance in athletes. This kind of training consists of alternating short bouts (10–30 s) of high-intensity exercise (>85–90% VO_2max_) interspersed with recovery periods [1,2] Repeated-sprint training (RST) is critical to improving endurance and performance in high-intensity exercise through peripheral muscular adaptations [3]. In this sense, RST has been one of the most important fitness indicators and has been widely studied [4,5]. RST combined with hypoxic conditions (RSH) has been proposed as an alternative to improve physical performance, achieving greater improvements than with sprint training alone [6,7].

Acute exercise in hypoxic conditions may induce a reduction in blood oxygen levels (hypoxemia) and may impact oxygen supply to tissues, leading to hypoxia (for example, decreased myoglobin oxygen saturation, and intramyocellular partial pressure) [8]. RSH results in greater heart rate, blood lactate, ventilation, and muscle deoxygenation [9]. However, long-term RSH training doubled the capacity to perform high-intensity workouts compared to training in normoxia [10]. Repeated sprint performance has been related, partially, to muscle reoxygenation capacity during rest and recovery periods. Muscle deoxygenation and the low rate of muscle oxygenation during recovery periods when executing RSH may greatly impact the ability to produce mechanical power in subsequent sprints [11,12]. Consequently, limited muscle reoxygenation capacity might compromise the metabolic recovery between sprints, resulting in a decreased repeated sprint performance. This evidence may suggest that muscle O_2_ availability is critical for repeated sprint performance [13]. This O_2_ requirement is exacerbated when the exercise is undertaken in hypoxia [13].

Athletes regularly undergo hypoxic training using various strategies [10,14], such as hypoxic chambers, masks, marquees, and sleep tents. In contrast to the traditional exposure to hypobaric hypoxia at altitude, some technological advances have led to the possibility of simulating hypoxia in normobaric settings. Recently, hypoxic devices have been developed that control critical environmental conditions such as temperature, humidity, and CO_2_ levels, reducing the side effects on performance due to these conditions [15]. Indeed, one of the main concerns in typical normobaric hypoxic systems is the high levels of CO_2_, the high temperature inside the tents, and the high relative humidity resulting from athletes’ exhalation and energy expenditure.

Combining reduced oxygen availability and increased environmental stressors may affect performance [16]. Therefore, we hypothesized that conditions inside a hypoxic tent, compared to those during mask-on training, could become hazardous and provoke a decrease in performance or modify the potential adaptations. We decided that this hypothesis should be explored because of the gap in the knowledge about the differences between a tent and mask-on sprint training [13]. Additionally, there is a lack of evidence on the potential adverse impacts of high environmental CO_2_ concentration, humidity, and temperature on performance during hypoxic and normoxic RST. Therefore, this study aimed to compare the environmental, perceptual, and external and internal workload acute effects of an RST session when performed inside a tent versus using a mask.

## 2. Methods

### 2.1. Participants

A total of twelve well-trained cyclists [17], were recruited to take part in the study (age 29 ± 9.8 years, weight 67 ± 9.7 kg, height 174.4 ± 6.6 cm, VO_2_max 70.3 ± 5.9 mL/kg/min, body fat 8.6 ± 4.2%). Participants were required to train for at least 15 h per week and be accustomed to high-intensity training. They also needed to report no previous acute hypoxic training (ascent over 1500 m) or mountain sickness in the past three months. There was no medication consumption during the trial, and no neuromuscular injuries in the past six months were reported. All participants were informed of the study’s aims, the protocol details, their rights during their participation, and potential risks caused by the experimental protocol. The study protocol followed the biomedical guidelines based on the Declaration of Helsinki (2013). Additionally, the protocol was reviewed and approved by the Institutional Review Board of the University of Extremadura (Reg. Code 174/2020).

### 2.2. Study Design

In a randomized, single-blinded, crossover protocol with a convenience sample, participants reported to the laboratory for a total of six sessions (one familiarization, one VO_2_max test, and four repeated-sprint testing sessions). The familiarization was performed using the same protocol for the RST sessions in both tent and mask-on conditions. The chronological execution of the four testing sessions was assigned randomly using an electronic random number generator. The four conditions were normoxia in a tent (RSN_Tent_) and mask-on (RSN_Mask_) and normobaric hypoxia in a tent (RSH_Tent_) and mask-on (RSH_Mask_) (see Figure 1).

Participants executed a warm-up of 9 min at moderate intensity with 100 watts and 80–90 rpm in each session, followed by a 1 min postblock with a 10 s submaximal sprint and a 50 s active recovery. The RSH consisted of three bouts of five series of 10 s repetitions (all-out sprints), with an active recovery of 20 s between repetitions and 180 s between series [17] (Figure 1). A final recovery of seven minutes was given for a total of 30 min per session. At least 72 h recovery was established between sessions, and all tests were performed at a similar time of the day to avoid circadian cycle effects. The protocol was performed in an isolated laboratory.

#### Hypoxic and Normoxic Stimulus

Each session was completed in a normobaric environment using a mask (RSN_Mask_ and RSH_Mask_) and a tent (RSN_Tent_ and RSH_Tent_). An altitude system (CAT-12, Louisville, Colorado) was used to simulate normoxia, at an inspired fraction of oxygen (FiO_2_) calibrated to 20.9% at 459 m.a.s.l., and hypoxia, at an FiO_2_ of 14.3%, corresponding to 3700 m.a.s.l. as at moderate to a high altitude [18] (Figure 1). A generator produced the hypoxia using a semipermeable filtration membrane (nitrogen filter technique) connected to a waterproof facial mask or a waterproof tent. The mask system included a neoprene harness that guaranteed ideal support for exercise in hypoxia. The tent (CAT-430, Louisville, CO, USA) used for both RSN_Tent_ and RSH_Tent_ conditions was 9 m^2^ in size, and for the RSN_Mask_ and RSH_Mask_ conditions, a 15 m^2^ room was used. In all conditions, only one person per test was allowed inside the area to avoid CO_2_ excess (Figure 2).

FiO_2_ was monitored constantly throughout the sessions using a handheld device (HANDI+, Maxtec, Salt Lake City, UT, USA). The participants’ hypoxic–normoxic blinding was achieved by covering all monitor displays; this was confirmed by interviewing the participants after the study. The participants were asked to avoid intense, prolonged, or strenuous exercise; alcohol consumption; and caffeine for at least 24 h before each session.

### 2.3. Material and Procedures

#### 2.3.1. Characterization

An electronic body weight scale (VitalControl, Hans Dinslage GmbH, Uttenweiler, Germany) was used to assess total body mass (kg) with a precision of 0.1 kg. Height was measured using a wall stadiometer. Fat percentage was estimated based on skinfold data of six regions (subscapular, triceps, suprailiac, abdomen, thigh, and gastrocnemius medialis) using special calipers (SlimGuide, Creative Health Products, Ann Arbor, MI, USA) with a precision of 0.2 mm. All anthropometric variables were measured following International Society for the Advancement of Kinanthropometry guidelines [18].

The VO_2_max was obtained using a gas analyser (Metalyzer 3b, CORTEX Biophysik GmbH, Leipzig, Germany) via an incremental test of 5 min warm-up at 50 W followed by 1 min rest. Then, participants cycled at 60 W, and the work rate was increased by 30 W every 3 min until exhaustion [19]. Achievement of VO_2_max was verified based on the incidence of the plateau phase (two successive maxima within 150 mL·min^−1^, averaging the data every 5 s) reached in VO_2_ [20].

#### 2.3.2. Indoor Environmental Conditions

The wet-globe bulb temperature (WGBT) and CO_2_ levels (ppm) were assessed using a digital logger (Green Eye, TechGrow, The Hague, The Netherlands) with an internal, stable nondispersive infrared sensor for CO_2_ detection (SenseAir^TM^, Delsbo, Sweden). WGBT was estimated using the indoor temperature (°C) and relative humidity (%) [21]. The equipment was calibrated via an automatic baseline calibration function and measured how much infrared light of a specific wavelength was absorbed by the surrounding air, which was then used to calculate CO_2_ concentration. The logger was placed one meter away, at the cyclist’s side, at a one-meter height. The external environmental conditions were controlled so that the training sessions would be carried out in similar situations (21–24 °C and 45–55% relative humidity), taking into account that throughout the sessions there would be changes in the store and the mask.

#### 2.3.3. Internal Load

Peak heart rate (bpm) for each sprint set was recorded using a heart rate monitor chest band (HRM-Tri, Garmin^TM^, Olathe, KS, USA) paired with a wrist band smartwatch (Forerunner 735xt, Garmin^TM^, Olathe, KS, USA) using ANT+ technology, with a sampling frequency of 4 Hz. This variable was obtained considering the peak heart rate from each set of the repeated-sprint workout. The heart rate strap was placed at xiphoid process height and adjusted using a strap system.

For the assessment of local muscle oxygen saturation (SmO_2_), a near-infrared spectroscopy device with a sampling frequency of 1 Hz (MOXY, Fortiori Design LLC, Minneapolis, MN, USA) was attached tightly to the right vastus lateralis muscle belly (~12 cm above the proximal lateral border of the patella) using a dark elastic strap to avoid light contamination and movement artifacts [13]. Skinfold thickness at the measurement point was measured using skinfold calipers to ensure that the skinfold thickness was less than half of the distance between the sensor´s emitter and the detector (~2.5 cm). Measurements were performed by an experienced investigator using a calibrated skinfold caliper. The investigator’s reliability was tested by performing two measurements on two consecutive days in the same site using an intraclass correlation statistic (ICC) and the standard error of measurement (SEM) (ICC = 0.998, SEM = 0.908 mm).

The vastus lateralis was selected based on previous evidence on its role in cycling [22]. The devices used for measurement were reliable, robust, and sensitive enough to detect haemoglobin changes during high-intensity actions [13,23,24]. The minimum percentage of SmO_2_ was measured for each sprint, and the mean for each sprint set was considered for analysis. The technology used allowed the assessment of oxyhaemoglobin/oxymyoglobin saturation (HbO_2_) from total haemoglobin/myoglobin (tHb) as follows:$${\mathrm{SmO}}_{2}=\frac{{\mathrm{HbO}}_{2}}{\mathrm{tHb}}$$

This was calculated by quantifying the variation in optical transmission by sequentially emitted light waves (630–850 nm) from light-emitting diodes into the tissue and recording the amount of light received. Using an algorithm, the system determined the amount of light absorbed at wavelengths corresponding to oxygenated and deoxygenated Hb using the Beer–Lambert law and tissue light propagation model processes.

The oxygen saturation (SpO_2_, %) was recorded using a finger oximeter (Checkme O_2_, Viatom Technology Co., Shenzhen, China). The average for each sprint set was considered for analysis.

Biochemical tests were performed using lactate and glucose handheld test meters after each sprint set. After cleaning and drying the area, two 0.2 μL capillary blood samples were obtained from the earlobe by micropuncture. Lactate concentration (BLa, mmol/L) was determined with a portable lactate test meter (Lactate Pro 2, Arkray Factory, Inc., Amstelveen, The Netherlands). Glucose concentration (mg/dL) was assessed using a portable electronic device (FreeStyle, Optium Neo, Oxon, UK). Reactive test strips were chosen as recommended by the manufacturers. Before each session, the equipment was calibrated following the manufacturers’ guidelines.

#### 2.3.4. External Load

Relative peak and mean power output (W/kg) and cadence (rpm) were measured, coupling the cassette of each participant’s bicycle to a cycle ergometer with electronic resistance (CycleOps^®^Hammer, Madison, WI, USA). The smart trainer assessed cadence and power with internal sensors that were paired to a smartwatch for future analysis (Forerunner 735xt, Garmin^TM^, Olathe, KS, USA).

#### 2.3.5. Perception

The rate of perceived exertion (RPE) was registered as a perceptual effort-fatigue marker. This variable indicates the subjective effort experienced by the athlete during and after an effort. The cyclists were accustomed to the RPE, assessed using the Borg Scale 0–10, where 0 was defined as “very, very light” effort and 10 as “maximum, strenuous” effort [25,26]. Perceived pain and respiratory difficulty were assessed using a visual analogue scale of 0–10 where 0 was defined as “minimal pain or difficulty” and 10 as “maximal pain or difficulty”. This perceptual assessment method has been used in other studies as a noninvasive technique for measuring the participants’ perceived degree of pain or discomfort [27].

### 2.4. Statistical Analysis

Results were reported using mean and standard deviation. The normality of the data was confirmed using the Shapiro–Wilk test. A mixed analysis of variance (MANOVA) was performed for each variable (4 × 4 or 4 × 3 as appropriate) to explore mean differences between groups. Three different *F* values were presented as an initial result for each test: an *F* value of the interaction between factors (*F*_Interaction_), an *F* value of the within-subjects factor (*F*_Timepoint_), and an *F* value of the between-subjects factor (*F*_Conditions_). Bonferroni correction was applied where necessary. Omega squared (*ω_p_*^2^) was used to qualify and quantify the magnitude of the differences (effect size: ES) as follows: <0.01, trivial; >0.01, small; >0.06, moderate; and >0.14, large [28]. The hypotheses were tested, setting alpha at *p* < 0.05. All data were analysed and systematized using the Statistical Package for the Social Sciences (SPSS, IBM, SPSS Statistics, v.22.0, Chicago, IL, USA).

## 3. Results

### 3.1. Indoor Environmental Conditions

The results suggested differences (*F*_interaction_ = 8.6, *p* < 0.01) in CO_2_ levels over time (*F*_Timepoint_ = 28.7, *p* < 0.01) and by condition (*F*_Condition_ = 46.9, *p* < 0.01) (Figure 3A). The CO_2_ levels increased significantly throughout the workout sets in all conditions (preworkout < 1st set < 2nd set < 3rd set, *p* < 0.01). Additionally, the CO_2_ levels when using the mask were lower (*p* < 0.01) than those inside the tent (RSH_Mask_ < RSH_Tent_, RSH_Mask_ < RSN_Tent_; RSN_Mask_ < RSH_Tent_, RSN_Mask_ < RSN_Tent_), independently of whether the workout was performed under hypoxic or normoxic conditions (*p* = 1, RSH_Mask_ = RSN_Mask_).

Differences in WGBT were found by time (*F*_Timepoint_ = 104.6, *p* < 0.01) and condition (*F*_Condition_ = 14.1, *p* < 0.01), with an additional effect of the interaction (*F*_interaction_ = 25.5, *p* < 0.01) (Figure 3B). The WGBT increased significantly throughout the workout sets in all conditions (preworkout < 1st set < 2nd set < 3rd set, *p* < 0.01). Additionally, WGBT was lower in (*p* < 0.01) the RSH_Mask_ and the RSN_Mask_ conditions than in the RSN_Tent_ condition, and lower in RSH_Mask_ than in RSH_Tent_. These results suggest that the WGBT when performing RSH or RSN in a tent was higher than in a mask.

### 3.2. Internal Load

There was no significant interaction in peak heart rate (*p* = 0.06). There were differences in peak heart rate over time, with a steady increase throughout the sprint sets (*p* = 0.01, 1st set < 2nd set < 3rd set), but no difference between conditions was found (*p* = 0.9) (Table 1). Regarding SmO_2_, there were no significant differences (*p* = 0.5) between conditions (*p* = 0.2) or product moment changes (*p* = 0.6) (Table 1).

In respect to SpO_2_, no differences were identified (*p*= 0.09) and no moment product differences (1st set = 2nd set = 3rd set) were found (*p* = 0.9). On the other hand, some differences were reported between conditions (*p* < 0.01) (Table 1). After Bonferroni correction, RSH_Tent_ presented lower SpO_2_ than the other three conditions (*p* < 0.05). Also, hypoxic conditions presented lower SpO_2_ than normoxic ones (*p* < 0.05).

There was a steady increase in BLa and glucose after each workout set (*p* < 0.01, preworkout < 1st set < 2nd set < 3rd set). However, there were no significant differences between conditions for either variable (*p* = 0.4) (Table 2).

### 3.3. External Load

No difference was found in peak power or mean power by condition (*p* = 0.7 and 0.8, respectively) or by time (*p* = 0.5 for both factors) (Table 3).

### 3.4. Perception

There were no moment product differences (*F*_Timepoint_ = 1.3, *p* = 0.3) or differences between conditions (*F*_Condition_ = 1.4, *p* = 0.3) in perceived respiratory difficulty or perceived pain (*F*_Condition_ = 1.4, *p* = 0.3 and *F*_Condition_ = 1.8, *p* = 0.2, respectively). In respect to RPE, there was a significant steady increase throughout the RSA series (*F*_Timepoint_ = 54.2, *p* < 0.01), preworkout < 1st set < 2nd set < 3rd set), but there were no differences by condition (*F*_Condition_ = 0.9, *p* = 0.4) (Figure 4).

## 4. Discussion

This study aimed to compare the environmental (CO_2_ and WGBT), perceptual (pain, respiratory difficulty and rate of perceived exertion), and external (peak and mean power output) and internal (peak heart rate, SmO_2_, SpO_2_, blood lactate and glucose) workload acute effects of an RST session when performed inside a tent versus using a mask in normoxic and normobaric–hypoxic conditions. The main results were that CO_2_ and WGBT levels increased steadily in all conditions and were lower when using a mask (RSN_Mask_ and RSH_Mask_) versus using a tent (RSH_Tent_ and RSN_Tent_). RSH_Tent_ presented lower SpO_2_ than the other three conditions, and hypoxic conditions presented lower SpO_2_ than normoxic ones. Peak heart rate, RPE, blood lactate, and glucose increased throughout the training as expected.

### 4.1. Indoor Environmental Conditions

In light of these results, during RST when using a tent, especially to simulate altitude, the cyclist’s exhalation and energy expenditure may provoke a rise in CO_2_ levels, relative humidity, and temperature (rise in WGBT), forming a closed circuit for the inspired and expired air. The rise in WGBT and CO_2_ was markedly higher in workouts performed in a tent than in mask-on sessions, especially at the end of the training (after 20–25 min), where CO_2_ levels were higher than 5000 ppm. Rebreathing of the expired air could lead to hypercapnic hypoxia, and in combination with normobaric hypoxia, this could increase acidity in the internal and external environment [29].

CO_2_ may be necessary when exercising in hypoxia to prevent and reduce hyperventilation-induced hypocapnia [30]. Certainly, there is still some grey area surrounding the adequate level of CO_2_ able to counteract hypoxia-related hypocapnia. Currently, the recommended level of CO_2_ is <3000 ppm when exercising [15]. Outdoor CO_2_ concentration is 350 ppm, and the indoor level is usually 700 ppm above outside levels, reaching 2800 ppm and provoking no severe symptoms [31]. The combined chronic effect of hypoxia and high CO_2_ during exercise must be explored in the future, based on recent evidence suggesting some issues when exposed to hypercapnic and hypoxic environments [32].

Future research should investigate how CO_2_ levels may impact physiological responses related to performance and health in the athlete. Methods may be explored to reduce hypocapnic conditions when exercising. For this purpose, some hypercapnic breathing techniques have been proposed as a strategy to reverse hypocapnia, but these methods are still debated [33]. Indeed, it is necessary to explore how CO_2_ levels could improve or decrease long-term performance. It is now known that physiological changes occur at CO_2_ exposure conditions between 500 and 5000 ppm, causing additional load to the respiratory system [34]. When exceeding 2000–5000 ppm of CO_2_, symptoms such as dizziness, tiredness, sleepiness, and headaches may be present, potentially altering human performance [35]. Inspiring CO_2_ may stimulate ventilation and acidify the blood, leading to a reduction in the affinity of haemoglobin for O_2_. This could affect O_2_ transportation, peak VO_2_max [36], respiratory alkalosis, increased blood lactate concentrations, early fatigue [37], and increases in heart rate and blood pressure. Finally, it may potentially compromise exercise performance.

Indoor CO_2_ rises during exercise because of human exhalation and perspiration and correlates with relative humidity increase [38]. Considering these results, athletes, coaches, medical staff, and other stakeholders should address the differences between performing RST using a tent and using a mask when simulating hypoxia. It should be known that repeated-sprint workouts represent a challenge to the body that is exacerbated under hypoxic conditions and could be boosted by other environmental factors such as high CO_2_ levels and the rise in WGBT [31,39]. Some studies have suggested that combined hypoxic and heat stress should be explored more in-depth [40,41] Despite the steady increase in WGBT inside the tent, the isolated and controlled conditions of this study did not allow the WGBT to rise to hazardous temperatures. However, the present results suggest that the WGBT must be monitored when hypoxia is applied in the field to avoid health issues. The possible increase in heat with a hypoxic stress factor does not improve performance, since it does not allow an adequate use of the energy substrate, and it instead has consequences on the athletes health [42]. In addition, studies have identified that walks in normobaric hypoxia tents increase the temperature because of sweating, which generates an increase in humidity and temperature [43]. This increase in temperature prevents thermoregulation at the neural level and can influence the thermal comfort zone. Therefore, decreased nerve conduction velocity is an important cause of the effects induced by store hypoxia on the functioning of the nervous system [44]. The added effect of CO_2_ levels may be studied in future research considering the intensity, volume, and type of workout.

### 4.2. Internal and External Load

It was found that SpO_2_ was lower in hypoxic conditions and lower when using a mask (RSN_Mask_ and RSH_Mask_) versus inside a tent (RSH_Tent_ and RSN_Tent_). In regular conditions, the body maintains stable SpO_2_ levels that remain almost unalterable during exercise. Despite the lack of O_2_ availability in hypoxic settings, the body manages to prevent SpO_2_ from falling below 90%. Still, it has been reported that during RSH, the SpO_2_ began to drop after 6 s sprints [45]. This evidence supports the present study’s findings, which showed a drop in SpO_2_ after the 10 s sprint set that was steady throughout the sets. A drop in SpO_2_ below 90% is known as hypoxemia and has been reported during RSH at simulated altitudes of 1800–2000 m.a.s.l and 3000 m.a.s.l [8]. Consequently, the combined effect of hypercapnic hypoxia and the rise in CO_2_ may partially explain the decrease in SpO_2_.

Contrary to the findings of some studies [8,46,47], in this study, the decrease in SpO_2_ and the possible decrease in oxyhaemoglobin during RSH did not impact SmO_2_. The difference in the SmO_2_ findings between studies could be due to the wide variety of measurement techniques [8] Considering that muscle oxygenation has an important role in skeletal muscle fatigue resistance, the maintenance of SmO_2_ evidenced in this study could also allow better cycling performance, resulting in a regular peak and mean power output. This response could be due to the high fitness level of the participants, who had strong neuromechanical adaptations that allowed them to perform independently of the adverse conditions (hypoxia and high CO_2_).

This could be caused by different physiological mechanisms (e.g., muscle recruitment or bioenergetic and O_2_ availability) that trigger a similar magnitude of peripheral fatigue in normoxic and hypoxic environments, despite the hypoxemia presented in hypoxia [48,49]. Indeed, this interaction between central and peripheral mechanisms regulating muscle fatigue during hypoxia is not yet well established [46]. For this reason, future research could focus on the mechanical and physiological muscle response during active recovery during sprints. It could be interesting to explore how muscle tissue reoxygenates in hypoxic and normoxic conditions.

Additionally, as expected, glucose availability, blood lactate, and heart rate increased throughout the RST. The greatest lactate accumulation and increase in glucose availability were in response to the 10 s sprints, which relied on the phosphagen system and glycolysis. Other studies indicated different results when analysing repeated sprints using 5 s [8] and 30 s [50] stimuli, one with a greater contribution of glycolysis compared to the phosphagen system.

### 4.3. Limitations

While this study intended to compare the environmental, perceptual, and external and internal workload acute effects of an RST session when performed inside a tent versus using a mask in normoxic and normobaric–hypoxic conditions, the outcomes of this trial must be seen in light of some limitations. These results were a reflection of acute adaptations only, and long-term studies are required. Likewise, the absence of arterial gas or HCO_3_ measurements prevented us from drawing firm conclusions about the impact of high levels of CO_2_ inside the tent compared to hypoxic conditions with a mask. This means that we could not take a position on the effect of CO_2_ levels on the response to hypo- and hypercapnia or how this condition could affect levels of load and perception. Considering that the results of this study were based on 10 s sprints, the outcomes cannot be extrapolated to other high-intensity activities with greater glycolytic or phosphagen components. Finally, a greater understanding of how the body responds during exercise and fatigue when exposed to conditions such as hypoxemia, hypercapnia, or hypocapnia is needed. In order to determine this, longer exposure to hypoxia and high CO_2_ is required.

## 5. Conclusions

In conclusion, in well-trained cyclists RSH could lead to acute conditions such as hypoxemia, which may be exacerbated when using a tent to simulate hypoxia compared to using a mask-based system, partly because of the high CO_2_ levels inside the tent. However, acute workout sessions based on repeated all-out sprints with a duration limited to 15 min (30 min whole session) did not seem to affect perceptual, external, or internal workload between hypoxic and normoxic conditions. This may suggest that exercise under hypoxia allows maintenance of physical and physiological performance, making it viable for sprint training.

## Figures and Tables

**Figure 1 jcm-10-04879-f001:**
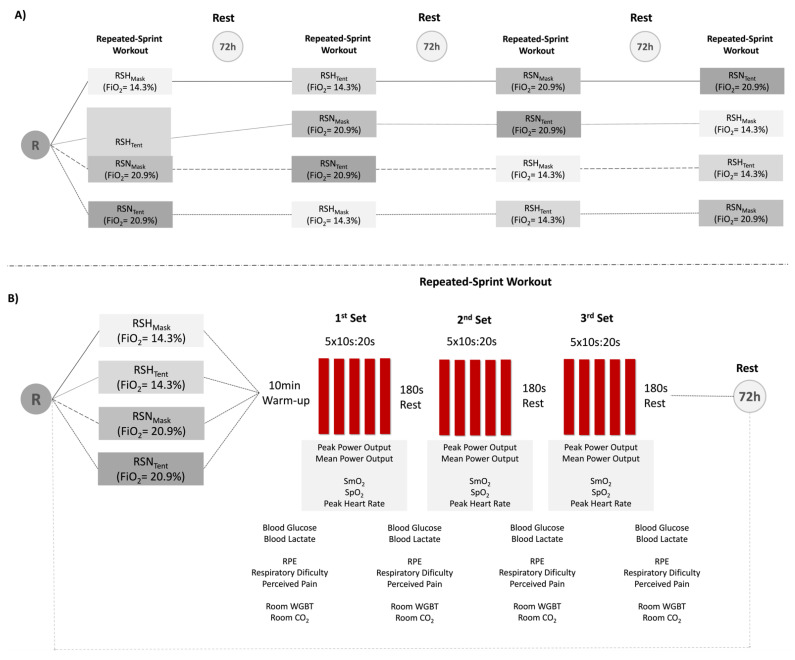
Schematic design of the test. (**A**) Randomization (R) of participants in each condition and flowchart of sessions. (**B**) Repeated-sprint test and chronological assessment of variables. RSH_Tent_ = repeated-sprint in normobaric hypoxia using a tent, RSH_Mask_ = repeated-sprint in normobaric hypoxia using a mask, RSN_Tent_ = repeated-sprint in normoxia using a tent, RSN_Mask_ = repeated-sprint in normoxia using a mask, FiO_2_ = inspired fraction of oxygen, SmO_2_ = muscle oxygen saturation, SpO_2_ = oxygen saturation, RPE = rate of perceived exertion.

**Figure 2 jcm-10-04879-f002:**
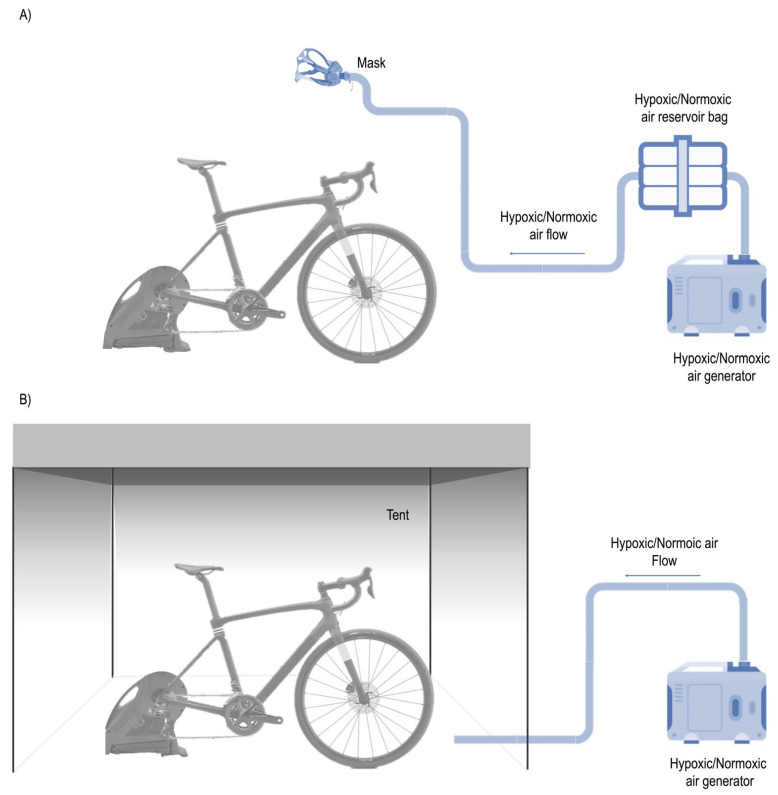
Training systems in normobaric hypoxia. (**A**) System setting for RSH_Mask_ = repeated-sprint in normobaric hypoxia using a mask and RSN_Mask_ = repeated-sprint in normoxia using a mask. (**B**) System setting for RSH_Tent_ = repeated-sprint in normobaric hypoxia and RSN_Tent_= repeated-sprint in normoxia using a tent.

**Figure 3 jcm-10-04879-f003:**
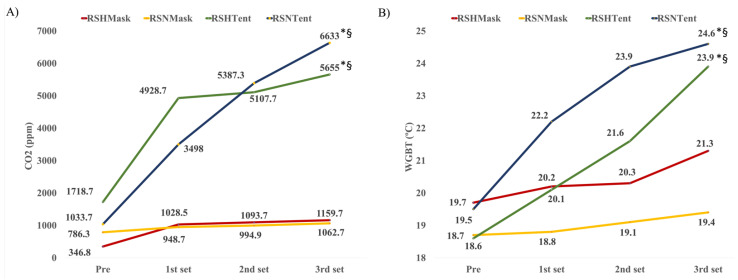
Changes in environmental conditions (**A**): CO_2_ and (**B**): WGBT); by time and conditions during a repeated-sprint workout in normoxia and hypoxia. Statistical difference between ^§^ RSNMask and * RSH_Mask_. RSHTent = repeated-sprint in normobaric hypoxia using a tent, RSHMask = repeated-sprint in normobaric hypoxia using a mask, RSNTent = repeated-sprint in normoxia using a tent, RSNMask = repeated-sprint in normoxia using a mask.

**Figure 4 jcm-10-04879-f004:**
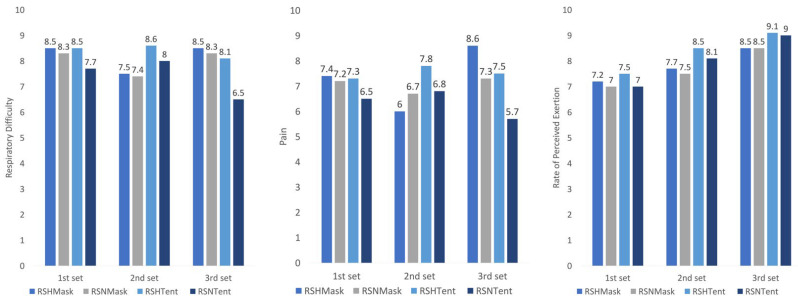
Changes in pain, exertion, and respiratory difficulty perception during a repeated-sprint workout in hypoxia and normoxia. RSH_Tent_ = repeated-sprint in normobaric hypoxia using a tent, RSH_Mask_ = repeated-sprint in normobaric hypoxia using a mask, RSN_Tent_ = repeated-sprint in normoxia using a tent, RSN_Mask_ = repeated-sprint in normoxia using a mask.

**Table 1 jcm-10-04879-t001:** Changes in cardiovascular responses due to repeated-sprint workout in normoxia and hypoxia.

Repeated-Sprint Workout	1st Set	2nd Set	3rd Set		
**Peak heart rate (bpm)**				*F*_timepoint_ (*p* value) ES—*ω_p_*^2^	
RSH_Mask_	163.5 ± 16.7	166.2 ± 14.6	167.9 ± 13.3	4.2 (0.01) 0.16, large	
RSN_Mask_	163.7 ± 13.7	165.8 ± 14.5	167.4 ± 13.5		
RSH_Tent_	166.1 ± 13.5	164.9 ± 11.6	164.1 ± 11.2		
RSN_Tent_	167.3 ± 15.3	168.3 ± 13.4	169.2 ± 13.1		
*F*_condition_ (*p* value) ES—*ω_p_*^2^	0.1 (0.9) 0, trivial			*F*_interaction_ (*p* value) ES—*ω_p_*^2^	2.1 (0.06)
**SmO_2_ (%)**				*F*_timepoint_ (*p* value) ES—*ω_p_*^2^	
RSH_Mask_	7.8 ± 2.6	8.2 ± 2.7	9.6 ± 4.4	1.5 (0.2) 0.03, small	
RSN_Mask_	11.1 ± 4.1	10.4 ± 5.9	10.7 ± 6.9		
RSH_Tent_	10.1 ± 3.7	9.3 ± 4.1	9.7 ± 4.7		
RSN_Tent_	11 ± 3.5	9.9 ± 4.2	10.4 ± 4.2		
*F*_condition_ (*p* value) ES—*ω_p_*^2^	0.6 (0.6) 0, trivial			*F*_interaction_ (*p* value) ES—*ω_p_*^2^	0.8 (0.5) 0, trivial
**SpO_2_ (%)**				*F*_timepoint_ (*p* value) ES—*ω_p_^2^*	
RSH_Mask_	88.2 ± 3.5	87.6 ± 4.3	86.8 ± 2.7	0.01 (0.9) 0, trivial	
RSN_Mask_	91.2 ± 4.9	92.5 ± 4.3	93.8 ± 3.1		
RSH_Tent_	84.3 ± 3.1	84.2 ± 3.7	83.2 ± 4.1		
RSN_Tent_	94.7 ± 1.9	93.8 ± 3.2	94.3 ± 2.4		
*F*_condition_ (*p* value) ES—*ω_p_*^2^	29.8 (*p* < 0.01) 0.6, large			*F*_interaction_ (*p* value) ES—*ω_p_*^2^	1.9 (0.09) 0.05, small

RSH_Tent_ = repeated-sprint in normobaric hypoxia using a tent, RSH_Mask_ = repeated-sprint in normobaric hypoxia using a mask, RSN_Tent_ = repeated-sprint in normoxia using a tent, RSN_Mask_ = repeated-sprint in normoxia using a mask, SmO_2_ = muscle oxygen saturation, SpO_2_ = oxygen saturation, ES = Effect Size, *ω_p_*^2^ = omega partial squared.

**Table 2 jcm-10-04879-t002:** Changes in biochemical responses due to repeated-sprint workout in normoxia and hypoxia.

Repeated-Sprint Workout	Pre	1st Set	2nd Set	3rd Set		
BLa (mmol/L)					*F*_timepoint_ (*p* value) ES, *ω_p_*^2^	
RSH_Mask_	1.4 ± 0.3	8.3 ± 2.7	12.5 ± 4.6	14.5 ± 3.5	285.3 (<0.01) 0.9, large	
RSN_Mask_	1.5 ± 0.4	6.9 ± 2.6	10.3 ± 3.8	12.1 ± 4		
RSH_Tent_	1.7 ± 0.4	9.4 ± 3.6	13.2 ± 4.3	14.6 ± 3.9		
RSN_Tent_	1.4 ± 0.4	7.5 ± 2.9	11.2 ± 3.4	12.9 ± 2.9		
*F*_condition_ (*p* value) ES—*ω_p_*^2^	1.4 (0.3) 0.02, small				*F*_interaction_ (*p* value) ES—*ω_p_*^2^	0.8 (0.5) 0.01, small
Glucose (mg/dL)					*F*_timepoint_ (*p* value) ES—*ω_p_^2^*	
RSH_Mask_	79.9 ± 25.5	82.4 ± 10.9	89.5 ± 11.1	100 ± 13.5	23.8 (<0.01) 0.55, large	
RSN_Mask_	89.6 ± 15.1	80 ± 12.4	89.7 ± 17.6	95.2 ± 16.4		
RSH_Tent_	82.9 ± 7.9	87.9 ± 9.2	103.1 ± 15.8	104.5 ± 10.7		
RSN_Tent_	84.7 ± 12	83.5 ± 11.4	91.1 ± 12	101 ± 14.9		
*F*_condition_ (*p* value) ES—*ω_p_*^2^	1.1 (0.4) 0.01, small				*F*_interaction_ (*p* value) ES—*ω_p_*^2^	1.5 (0.1) 0.03, small

RSH_Tent_ = repeated-sprint in normobaric hypoxia using a tent, RSH_Mask_ = repeated-sprint in normobaric hypoxia using a mask, RSN_Tent_ = repeated-sprint in normoxia using a tent, RSN_Mask_ = repeated-sprint in normoxia using a mask, BLa = blood lactate, ES = Effect Size, *ω_p_*^2^ = omega partial squared.

**Table 3 jcm-10-04879-t003:** Differences in power output by condition during RSA.

Repeated-Sprint Workout	1st Set	2nd Set	3rd Set		
**Peak power (w/kg)**				*F*_timepoint_ (*p* value) ES—*ω_p_*^2^	
RSH_Mask_	9.9 ± 3.2	10.1 ± 3	10.1 ± 3.1	0.7 (0.5) 0, trivial	
RSN_Mask_	10.3 ± 3.2	10.3 ± 3.1	10.5 ± 3.3		
RSH_Tent_	9.5 ± 3.3	9.4 ± 3.4	9.1 ± 3.3		
RSN_Tent_	10.7 ± 3.4	10.7 ± 3.2	10.5 ± 3.1		
*F*_condition_ (*p* value) ES—*ω_p_*^2^	0.5 (0.8) 0, trivial			*F*_interaction_ (*p* value) ES- *ω_p_*^2^	0.7 (0.6) 0, trivial
**Mean power (w/kg)**				*F*_timepoint_ (*p* value) ES—*ω_p_*^2^	
RSH_Mask_	7.4 ± 2.6	7.9 ± 2.6	7.7 ± 2.4	0.5 (0.5) 0, trivial	
RSN_Mask_	7.9 ± 2.4	7.9 ± 2.3	8 ± 2.7		
RSH_Tent_	7.2 ± 2.4	7.2 ± 2.6	6.8 ± 2.4		
RSN_Tent_	8.2 ± 2.5	8.2 ± 2.3	8.1 ± 2.7		
*F*_condition_ (*p* value) ES—*ω_p_*^2^	0.4 (0.7) 0, trivial			*F*_interaction_ (*p* value) ES—*ω_p_*^2^	0.9 (0.5) 0, trivial

RSH_Tent_ = repeated-sprint in normobaric hypoxia using a tent, RSH_Mask_ = repeated-sprint in normobaric hypoxia using a mask, RSN_Tent_ = repeated-sprint in normoxia using a tent, RSN_Mask_ = repeated-sprint in normoxia using a mask, ES = Effect Size, *ω_p_*^2^ = omega partial squared.

## Data Availability

No applicable.

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
