# Peer review of "Tent versus Mask-On Acute Effects during Repeated-Sprint Training in Normobaric Hypoxia and Normoxia"

_jcm, 2021, doi:10.3390/jcm10214879_

Round 1

Reviewer 1 Report

The study investigated the effects of RST using different hypoxia conditions. The study is quite interesting and the authors conducted and discussed most of the results correctly. Some minor issues should be addressed before the formal acceptance of the manuscript.

Abstrat

  • Please add full words before acronyms throughout the text.
  • Please add effect sizes for significants results, it is important for results interpretation.
  • Peak heart rate, RPE, blood lactate .... increased .. Also add p values and effect sizes for these parameters

Introduction

The introduction is well written, some minors issues to be corrected.

  • L36: i think here you mean high-intensity sports' performance, and please add reference
  • How RST is a performance indicators? Please reword it
  • L47 : compared to normoxic training 
  • L53: add reference
  • L66: the conditions inside a hypoxic....
  • Adding hypotheses at the end of the introdution would be very beneficial

Methods

Participants

  • If i understand well, only 4 athletes were recruited in this study ? i think the sample size is very small and statistical power analysis is needed to justify the sample size used.

2.3.1 Characterisation

Please add references supporting the incremental test used here.

Statistical analysis and Results

Please add effect size's values for differences found.

Discussion

  • L226- 275: Please delete p values
  • Authors are called to bring more explanations on how conditions altered environmental conditions.
  • L332: Please reword it to avoid repetition ( studies; in this study).

Author Response

First of all, we would like to thank you again for this second review and your comments in order to improve the article.

We have made some changes in the document according to your suggestions. Every point has been discussed. Changes made in the document are written in "Red" and the responses are “written in blue” within this file. You can also view the control of changes in the manuscript. We hope that our responses can satisfy your comments.

reviewer 1

General comments:

The study investigated the effects of RST using different hypoxia conditions. The study is quite interesting and the authors conducted and discussed most of the results correctly. Some minor issues should be addressed before the formal acceptance of the manuscript.

Specific comments:

Abstract

    Please add full words before acronyms throughout the text.

    Please add effect sizes for significants results, it is important for results interpretation.

    Peak heart rate, RPE, blood lactate .... increased .. Also add p values and effect sizes for these parameters

Response: Thanks for the comments. Therefore, we have added the acronyms to the abstract and performed the effect size analysis.

Introduction

The introduction is well written, some minors issues to be corrected.

    L36: i think here you mean high-intensity sports' performance, and please add reference

    How RST is a performance indicators? Please reword it

Response: we have re-inscribed the phrase to be more clear: “improving endurance and performance in high intensity exercise..”

Reference

Faiss, R., Léger, B., Vesin, J. M., Fournier, P. E., Eggel, Y., Dériaz, O., & Millet, G. P. (2013). Significant molecular and systemic adaptations after repeated sprint training in hypoxia. PloS one, 8(2), e56522.

    L47 : compared to normoxic training

Response: we have re-inscribed the phrase: “Compared to training in normoxia”

    L53: add reference

Response: added reference

Billaut, F., & Buchheit, M. (2013). Repeated‐sprint performance and vastus lateralis oxygenation: Effect of limited O 2 availability. Scandinavian journal of medicine & science in sports, 23(3), e185-e193.

    L66: the conditions inside a hypoxic....

    Adding hypotheses at the end of the introdution would be very beneficial

Response: we have added this phrase: “Therefore, we hypothesized that.” In the line 66

Methods

Participants

    If i understand well, only 4 athletes were recruited in this study ? i think the sample size is very small and statistical power analysis is needed to justify the sample size used.

Response:  No, the sample consisted of 12 well-trained athletes. Level 4 of performance refers to athletes with VO2max > 65, in any of the cases we provide the statistical power of the results.

2.3.1 Characterisation

Please add references supporting the incremental test used here.

Response: added reference

Beaver, W. L., Wasserman, K. A. R. L. M. A. N., & Whipp, B. J. (1986). A new method for detecting anaerobic threshold by gas exchange. Journal of applied physiology, 60(6), 2020-2027.

Statistical analysis and Results

Please add effect size's values for differences found.

Response: We have added to the text the information on how the effect size was obtained and the ranking to consider it qualification in the corresponding section.

Discussion

  • L226- 275: Please delete p values
  • Authors are called to bring more explanations on how conditions altered environmental conditions.
  • L332: Please reword it to avoid repetition ( studies; in this study).

Response: We have removed the p values and removed the study words throughout the discussion

And we add this phrase to the environmental conditions:

The possible increase in heat with a hypoxic stress factor does not improve performance since it does not allow an adequate use of the energy substrate and rather has consequences on the athletes health (Rendell et al., 2017). In addition, studies have identified that walks in normobaric hypoxia tents increase the temperature due to sweating, which generates an increase in humidity and temperature (Richardson et al., 2008). The increase in temperature prevents thermoregulation at the neural level and can influence the thermal comfort zone. therefore, decreased nerve conduction velocity is an important cause of the effects induced by store hypoxia on the functioning of the nervous system (Malanda et al., 2008).

Reference:

Rendell, R. A., Prout, J., Costello, J. T., Massey, H. C., Tipton, M. J., Young, J. S., & Corbett, J. (2017). Effects of 10 days of separate heat and hypoxic exposure on heat acclimation and temperate exercise performance. American Journal of Physiology-Regulatory, Integrative and Comparative Physiology, 313(3), R191-R201.

Richardson, A., Twomey, R., Watt, P., & Maxwell, N. (2008). Physiological responses to graded acute normobaric hypoxia using an intermittent walking protocol. Wilderness & environmental medicine, 19(4), 252-260.

Malanda, U. L., Reulen, J. P. H., Saris, W. H. M., & van Marken Lichtenbelt, W. D. (2008). Hypoxia induces no change in cutaneous thresholds for warmth and cold sensation. European journal of applied physiology, 104(2), 375-381.

We appreciate your comments!

Reviewer 2 Report

This study aimed to compare the environmental (CO2 and WGBT), perceptual (pain, respiratory difficulty and rate of perceived exertion), and external (peak and mean power output) and internal (peak heart rate, SmO2, SpO2, blood lactate and glucose) workload acute effects of an RST session when performed inside a tent versus using a mask in normoxic and normobaric-hypoxic conditions. Authors indicated that sprint in hypoxic tent
 presented lower SpO2 than the other three conditions.
Overall study is well written and the data are clearly presented. I have only few comments I believe  will improve this study:
Please explain any abbreviations in the abstract or preferably do not use abbreviations.
L97: please describe the loads: how they were chosen, how much they were, etc. What was the intensity in warm up?
L142: Please prove that VO2max was achieved but not VO2peak - What criteria were used to determine VO2max?

Author Response

First of all, we would like to thank you again for this second review and your comments in order to improve the article.

We have made some changes in the document according to your suggestions. Every point has been discussed. Changes made in the document are written in "Red" and the responses are “written in blue” within this file. You can also view the control of changes in the manuscript. We hope that our responses can satisfy your comments.

Reviewer 2

This study aimed to compare the environmental (CO2 and WGBT), perceptual (pain, respiratory difficulty and rate of perceived exertion), and external (peak and mean power output) and internal (peak heart rate, SmO2, SpO2, blood lactate and glucose) workload acute effects of an RST session when performed inside a tent versus using a mask in normoxic and normobaric-hypoxic conditions. Authors indicated that sprint in hypoxic tent

 presented lower SpO2 than the other three conditions.

Overall study is well written and the data are clearly presented. I have only few comments I believe  will improve this study:

Please explain any abbreviations in the abstract or preferably do not use abbreviations.

L97: please describe the loads: how they were chosen, how much they were, etc. What was the intensity in warm up?

Response: we have added details of the warm-up: “participants executed a warm-up of 9 minutes at moderate intensity with 100 watts and 80-90 rpm, followed by a 1-minute post-block with a 10-second submaximal sprint and a 50-second active recovery. The RSH consisted of three..”

L142: Please prove that VO2max was achieved but not VO2peak - What criteria were used to determine VO2max?

Response: we have added this phrase: “Achievement of VO2max was verified based on the incidence of the plateau phase (two successive maximal within 150 mL·min−1, averaging the data every 5 s) reached in VO2 (Lacour et al., 1991)”.

Lacour JR, Padilla-Magunacelaya S, Chatard JC, Arsac L, Barthélémy JC. Assessment of running velocity at maximal oxygen uptake. Eur J Appl Physiol Occup Physiol 1991 622 [Internet]. 1991 Mar [cited 2021 Jul 30];62(2):77–82. Available from: https://link.springer.com/article/10.1007/BF00626760

Reviewer 3 Report

Major comments:
- The manuscript is written in high-quality English and the flow of reading is very good. Only minor errors and glitches were detected (e.g., a period missing, an extra space, stuff like that), which could benefit from an overall review of the manuscript.
- The introduction establishes a very easy-to-follow rationale that wastes no time. The authors go straight to the point and provide a very comprehensible rationale that properly justifies performing this particular experiment.
- The type of study should have been pre-registered in a platform such as clinicaltrials.gov or similar. However, since there is a protocol approved by the IRB of the authors’ university, with a specific code (174/2020), I advise the authors to provide that protocol as supplementary material, even if the original language is not English.
- The randomized crossover design is appropriate, and the familiarisation session helps in reducing a learning effect, therefore making the results more trustworthy. The randomization process is properly described. Although there is no statement on whether the allocation sequence was concealed, I do not believe this to be such a key feature in a crossover trial.
- The minimum 72 h wash-out period between sessions is likely to have been enough for avoiding carry-over effects, which again speaks to good practices and avoids statistical complexities.
- Given the standardization reported in the methods, I’m guessing that room temperature and humidity were also controlled for, but it would be nice if the authors explicitly addressed it.
- The global description of the procedures is very clear and easy to follow.
- The authors should provide information concerning the Typical Error of Measurement for the handheld HANDI+ device, as well as for the digital logger used to assess WGBT, the HR monitor chest band, and other devices used in this experiment. This is important for interpreting the results, i.e., to understand if they surpass the error margin of the device and therefore constitute “true” results. In absence of data on T.E.M., please report known ICCs for each device.
- The blinding was a nice addition to this protocol, even if the nature of the tests would likely not be all that influenced by knowledge of the intervention. Still, it represents a nice touch.
- I have concerns with skinfold data, though. Please, provide details concerning the number of assessors, their background/training in measuring skinfolds, and provide values for intra- and/or inter-assessors reliability.
- Regarding the sample, the authors should also provide additional information, such as: (i) Was there any a priori calculation of power, or was this a convenience sample? (ii) If this was a convenience sample, state it explicitly and please provide an a posteriori power analysis, to help to frame the results; (iii) Were the 12 cyclists the totality of the initially recruited sample, or a subset of a larger initial sample? (iv) If they were a subset, what happened to the other participants? Please, clarify these issues.
- Please state whether the cyclists were accustomed to RPE. I’m guessing they were, but please address this explicitly.
- One major shortcoming pertaining to the statistical analysis is the lack of calculation of effect sizes. How big were the differences between conditions? Please add these calculations, and use their values to improve the framing of the results and discussion.
- The results are presented in a clear manner, but again there is no knowledge of the magnitude of effects. Please report effect sizes (e.g., partial eta-squared).
- In line with the previous section, the discussion is very clear-cut and informative. However, I believe there should be a section specifically devoted to the limitations of the study, near the end of the discussion. A major limitation that should be acknowledged more explicitly is that these results are a reflection of acute adaptations only, and long-term studies are required. In general, the entire penultimate paragraph of the discussion could be separated from it, constituting a limitations section. This would make it easier for readers to separate the general discussion from the limitations. Also, the final paragraph should appear in a separate, final section titled “Conclusion”.

Overall, I congratulate the authors for the relevance of the theme, general methodological quality, and quality of writing and presentation. However, additional information and clarifications are required in the methods, and effect sizes should be calculated and used to more completely frame the results.

Additional details specifically concerning the abstract:
- Replace WGBT with the complete expression “wet-globe bulb temperature” in its first appearance.
- Please reframe the level of the cyclists, as the meaning of “level four” is not clear for those not involved in that particular sport. In the body of the manuscript, “level four” can be explained and used, but in the abstract, I feel it does not convey an appropriate notion of their competitive level.
- There is an “x” missing in “VO2ma”.
- Use indices (i.e., subscript) for the numbers used in specific abbreviations or chemical symbols (e.g., for CO2 and VO2max). Since the manuscript was likely prepared with Microsoft Word, it is easy to format those numbers as indices (i.e., subscript).

Author Response

First of all, we would like to thank you again for this second review and your comments in order to improve the article.

We have made some changes in the document according to your suggestions. Every point has been discussed. Changes made in the document are written in "Red" and the responses are “written in blue” within this file. You can also view the control of changes in the manuscript. We hope that our responses can satisfy your comments.

Reviewer 3

Major comments:
- The manuscript is written in high-quality English and the flow of reading is very good. Only minor errors and glitches were detected (e.g., a period missing, an extra space, stuff like that), which could benefit from an overall review of the manuscript.
- The introduction establishes a very easy-to-follow rationale that wastes no time. The authors go straight to the point and provide a very comprehensible rationale that properly justifies performing this particular experiment.

Response: We appreciate your comments!

- The type of study should have been pre-registered in a platform such as clinicaltrials.gov or similar. However, since there is a protocol approved by the IRB of the authors’ university, with a specific code (174/2020), I advise the authors to provide that protocol as supplementary material, even if the original language is not English.

Response: we provide the protocol approved by the bioethics committee as supplementary material.

- The randomized crossover design is appropriate, and the familiarisation session helps in reducing a learning effect, therefore making the results more trustworthy. The randomization process is properly described. Although there is no statement on whether the allocation sequence was concealed, I do not believe this to be such a key feature in a crossover trial.

- The minimum 72 h wash-out period between sessions is likely to have been enough for avoiding carry-over effects, which again speaks to good practices and avoids statistical complexities.

Response: we appreciate your comment according to the study design

- Given the standardization reported in the methods, I’m guessing that room temperature and humidity were also controlled for, but it would be nice if the authors explicitly addressed it.

Response: thank you for your observation, we have addressed this matter with a new phrase: “The external environmental conditions were controlled so that the training sessions would be carried out in similar conditions (21-24 ºC and 45-55% relative humidity), taking into account that throughout the sessions there would be changes in the store and the mask”

- The global description of the procedures is very clear and easy to follow.
- The authors should provide information concerning the Typical Error of Measurement for the handheld HANDI+ device, as well as for the digital logger used to assess WGBT, the HR monitor chest band, and other devices used in this experiment. This is important for interpreting the results, i.e., to understand if they surpass the error margin of the device and therefore constitute “true” results. In absence of data on T.E.M., please report known ICCs for each device.

Response: we have provided the ICCS of each of the devices

- The blinding was a nice addition to this protocol, even if the nature of the tests would likely not be all that influenced by knowledge of the intervention. Still, it represents a nice touch.

Response: We appreciate your comment

- I have concerns with skinfold data, though. Please, provide details concerning the number of assessors, their background/training in measuring skinfolds, and provide values for intra- and/or inter-assessors reliability.

Response: We have added the required information in the SmO2 measurement section (Internal load):

 “and they were performed by an experienced investigator using a calibrated Harpenden caliper (John Bull, England). The investigator’s reliability was tested by performing 2 measurements on 2 consecutive days in the same site using an intraclass correlation statistic (ICC) and the standard error of measurement (SEM) (ICC = 0.998, SEM = 0.908 mm).”

- Regarding the sample, the authors should also provide additional information, such as: (i) Was there any a priori calculation of power, or was this a convenience sample? (ii) If this was a convenience sample, state it explicitly and please provide an a posteriori power analysis, to help to frame the results; (iii) Were the 12 cyclists the totality of the initially recruited sample, or a subset of a larger initial sample? (iv) If they were a subset, what happened to the other participants? Please, clarify these issues.

Response:

Thanks for the recommendations provided by the reviewer, below you address the questions:

  1. i) It was a convenience sample (added to the text, in the methodology and the study design)
  2. ii) We have performed the statistical power analysis

iii) The entire sample consisted of 12 cyclists.

- Please state whether the cyclists were accustomed to RPE. I’m guessing they were, but please address this explicitly.

Response: We have added this information in the Perception section: The cyclists were accustomed to the RPE, assessed using..”

- One major shortcoming pertaining to the statistical analysis is the lack of calculation of effect sizes. How big were the differences between conditions? Please add these calculations, and use their values to improve the framing of the results and discussion.
- The results are presented in a clear manner, but again there is no knowledge of the magnitude of effects. Please report effect sizes (e.g., partial eta-squared).

Response: we have calculated and added the effect size in the tables and results of the study. We have used omega partial squared due to sample size.

- In line with the previous section, the discussion is very clear-cut and informative. However, I believe there should be a section specifically devoted to the limitations of the study, near the end of the discussion. A major limitation that should be acknowledged more explicitly is that these results are a reflection of acute adaptations only, and long-term studies are required. In general, the entire penultimate paragraph of the discussion could be separated from it, constituting a limitations section. This would make it easier for readers to separate the general discussion from the limitations. Also, the final paragraph should appear in a separate, final section titled “Conclusion”.

Response:

We have added the phrase “These results are a reflection of acute adaptations only, and long-term studies are re-quired. Likewise ”.. at the beginning of the limitations.

Likewise, we have separated the limitations and conclusions

Overall, I congratulate the authors for the relevance of the theme, general methodological quality, and quality of writing and presentation. However, additional information and clarifications are required in the methods, and effect sizes should be calculated and used to more completely frame the results.

Response: Thank you for the review, your recommendations and that the study met your expectations.

Additional details specifically concerning the abstract:
- Replace WGBT with the complete expression “wet-globe bulb temperature” in its first appearance.
- Please reframe the level of the cyclists, as the meaning of “level four” is not clear for those not involved in that particular sport. In the body of the manuscript, “level four” can be explained and used, but in the abstract, I feel it does not convey an appropriate notion of their competitive level.
- There is an “x” missing in “VO2ma”.
- Use indices (i.e., subscript) for the numbers used in specific abbreviations or chemical symbols (e.g., for CO2 and VO2max). Since the manuscript was likely prepared with Microsoft Word, it is easy to format those numbers as indices (i.e., subscript).

Response: Thank you for your recommendations, we have addressed each of these details in the new text.
